# Research on Indoor 3D Positioning Algorithm Based on WiFi Fingerprint

**DOI:** 10.3390/s23010153

**Published:** 2022-12-23

**Authors:** Lixing Wang, Shuang Shang, Zhenning Wu

**Affiliations:** 1School of Computers and Engineering, Northeastern University, Shenyang 110000, China; 2College of Information Science and Engineering, Northeastern University, Shenyang 110000, China

**Keywords:** WiFi fingerprinting, indoor positioning, DNN, TCN

## Abstract

Indoor 3D positioning is useful in multistory buildings, such as shopping malls, libraries, and airports. This study focuses on indoor 3D positioning using wireless access points (AP) in an environment without adding additional hardware facilities in large-scale complex places. The integration of a deep learning algorithm into indoor 3D positioning is studied, and a 3D dynamic positioning model based on temporal fingerprints is proposed. In contrast to the traditional positioning models with a single input, the proposed method uses a sliding time window to build a temporal fingerprint chip as the input of the positioning model to provide abundant information for positioning. Temporal information can be used to distinguish locations with similar fingerprint vectors and to improve the accuracy and robustness of positioning. Moreover, deep learning has been applied for the automatic extraction of spatiotemporal features. A temporal convolutional network (TCN) feature extractor is proposed in this paper, which adopts a causal convolution mechanism, dilated convolution mechanism, and residual connection mechanism and is not limited by the size of the convolution kernel. It is capable of learning hidden information and spatiotemporal relationships from the input features and the extracted spatiotemporal features are connected with a deep neural network (DNN) regressor to fit the complex nonlinear mapping relationship between the features and position coordinates to estimate the 3D position coordinates of the target. Finally, an open-source public dataset was used to verify the performance of the localization algorithm. Experimental results demonstrated the effectiveness of the proposed positioning model and a comparison between the proposed model and existing models proved that the proposed model can provide more accurate three-dimensional position coordinates.

## 1. Introduction

Since the concept of Industry 4.0 was proposed, location-based services (LBS) have attracted significant attention because of their social and commercial value [1] and the technology behind these services has undergone advancement in the field of outdoor positioning. The global navigation satellite system (GNSS), which occupies a dominant position in outdoor positioning [2], has been put into use, bringing great convenience to people’s daily lives; however, people generally tend to spend 87–90% of their time indoors [3]. Indoor positioning has broad application prospects, such as guiding customers to shops and food courts in shopping malls, and pushing store information. During an emergency rescue, first responders can use indoor positioning to quickly locate a person in distress, and in terms of personnel and goods management, real-time monitoring of personnel status in chemical plants can be done to avoid factory accidents. In hospitals, indoor positioning is used to provide consultation services and push public service information and in museums, accurate indoor positioning can convert a user’s phone into a virtual guide. Although GNSS has been able to more accurately realize outdoor positioning, it is limited by several factors in indoor scene positioning applications. Satellite signals from the outdoor environment to the indoor environment will suffer from serious attenuation or even complete blockage [4] which in turn causes the devices to receive a very weak signal, and therefore, an inaccurate location is obtained. Indoor positioning remains a problem that needs to be solved, especially in large multistory buildings.

Industries and academia have devoted themselves to research on indoor positioning technology and methods. In the past decade, the number of papers published in the field of indoor positioning has rapidly increased and many wireless technologies have been studied for indoor positioning, including ultrasonic, Bluetooth, radio frequency identification (RFID), ultra-wide bandwidth (UWB), and WiFi technologies. The advantages, disadvantages, and positioning accuracies of these techniques are summarized in Table 1 [5].

Among these technologies, ultrasonic, RFID, UWB, Bluetooth, infrared, and ZigBee indoor positioning techniques require specific hardware support and several positioning beacons, such as infrared, ultrasonic, or RFID beacon whilst WiFi is the most widely deployed indoor wireless infrastructure [20]. The Wi-Fi positioning technique does not require additional hardware and can provide indoor positioning solely using an evenly distributed wireless network set up in the building and the smart device at the user’s end. Owing to the increasing popularity of WiFi, WiFi signals are now ubiquitous in society, as they meet the infrastructure requirements of WiFi positioning. In addition, many people have become habituated to using Wi-Fi. Consequently, WiFi positioning has a substantial advantage over other indoor positioning techniques. Therefore, this study focuses on the WiFi fingerprint indoor positioning technology and aims to design an indoor 3D positioning algorithm. First, it constructs temporal fingerprint slices based on sliding windows. The algorithm uses temporal fingerprint slices as inputs. Temporal features are considered and compared with the normal positioning method. In the proposed algorithm, a temporal convolutional network (TCN) feature extractor is used for the automatic extraction of spatiotemporal features to achieve end-to-end indoor 3D positioning.

The remainder of this paper is organized as follows. Section 2 briefly reviews previous research on WiFi fingerprint-based indoor-positioning methods and deep learning. Section 3 describes the framework of the proposed 3D temporal fingerprint algorithm. Section 4 presents a test of the proposed algorithm and presents an analysis of the results. Finally, Section 5 presents conclusions and provides an outlook for future research.

## 2. Literature Review

The fingerprint positioning method must construct a fingerprint map in advance from the target area [21] and a popular method is to construct fingerprint maps in a discrete manner. This approach partitions discrete predefined reference points into an unlocatable environment. The fingerprint is randomly collected at the reference points and did not contain temporal information. Many machine learning algorithms, such as K-nearest neighbor (KNN) [22], naive Bayes [23], support vector machine [24], and random forest [25], are used for plane position estimation. RADAR is the first indoor positioning system to use the location-fingerprint method. This method uses the K-nearest neighbor algorithm, with the core idea being to use the Euclidean distance to measure the similarity of location fingerprints. At present, the improvement of the K-nearest neighbor algorithm in fingerprint-based indoor positioning remains an important research direction. Brunato proposed the weighted K-nearest neighbor (WKNN) positioning algorithm, which assigns different weights to K reference points according to the size of similarity and obtains the location coordinate estimation of the target by calculating the weighted average of coordinates of K neighboring points [26]. Li et al. used Manhattan distance as a similarity measure, which, compared to the Euclidean distance-based WKNN algorithm, improved the accuracy by 33.82% [27]. Tran and Pham proposed a positioning model based on support vector machine (SVM) classification, and found that SVM parameters have a significant impact on positioning performance [24]. Li et al. proposed an indoor positioning prediction model based on an SVM and used an improved sparrow search algorithm to optimize the parameters. The effect of ensemble learning is better than that of a single learner, which is introduced into indoor positioning to further improve positioning accuracy [28]. For example, Lee et al. proposed an indoor positioning model based on random forest classification. With the emergence of deep learning, researchers are attempting to apply deep learning to indoor positioning to achieve better positioning results [29]. In 2016, Zhang et al. introduced deep learning into Wi-Fi fingerprint positioning for the first time. The authors proposed a four-layer deep neural network positioning model that divided the indoor environment into hundreds of square grids, and the number of neurons in the output layer of the neural network classifier was equal to the number of grids [30]. Felix et al. applied deep belief networks and two different types of restricted Boltzmann machines for indoor positioning [31].

For the study of 3D positioning, Sospedra et al. proposed the first open-source multifloor location fingerprint dataset and adopted the KNN method [32]. Nguyen summarized and compared mainstream WiFi fingerprint-based positioning algorithms such as WKNN, naive Bayes, and neural networks, with experimental results showing that the WKNN algorithm performed well in most cases. In 3D positioning, Clustering is a popular method for floor recognition [33]. Zhou et al. applied the k-means algorithm to cluster a location fingerprint map. The cluster center was used to represent different floors, and the cluster center with the largest similarity to the undetermined site was selected to determine the floor. The selection of the cluster center and K value had a significant influence on the results [34]. Cramariuc et al. proposed an affinity propagation clustering algorithm based on the penalized log-Gaussian distance to perform affinity propagation clustering on fingerprint maps, which improved the performance of fingerprint clustering in multistory buildings. However, the fingerprint data of different layers were still mistakenly assigned to the same cluster [35]. Kim et al. proposed a 3D positioning architecture for an autoencoder combined with a deep neural network (DNN) classifier. The authors used the autoencoder to reduce the dimension of input data, achieving a floor hit ratio of 91.27% and a position coordinate positioning accuracy of 9.29 m [36]. Song et al. proposed different models to predict buildings, floors, and coordinates. Given that buildings are far away from each other, a fully connected neural network was used for building prediction, and a combination of a fully connected neural network and a one-dimensional convolutional neural network was utilized for floor prediction and coordinate prediction, respectively [37].

Another method for constructing fingerprint maps is to use continuous data. This approach involves collecting fingerprints along the trajectory sequentially and recording the location information of the sampling points. The collected fingerprints contained temporal information and the historical information of the walking trajectory can be used to enhance position estimation. Au et al. used a Kalman filter for continuous localization which requires the assumption of Gaussian noise distribution and linear motion [38]. Hoang proposed the SRL (Simultaneous Region Localization)-KNN positioning algorithm, which does not require additional assumptions, and combines the historical location information of the user into the KNN to restrict the search space of the nearest neighbor nodes to the circle. The estimated position at the previous time point was at the center of the circle, the radius is determined by the user’s moving speed and the time difference between two consecutive measurements to reduce spatial ambiguity in positioning [39]. Bai et al. applied deep learning to continuous positioning and proposed a positioning model based on a double-layer cascaded RNN. The first RNN layer was used for position matching to match the historical fingerprint measurements with the corresponding estimated position and the second RNN layer is used for position filtering. The output of the first RNN layer is used as the input for the second RNN layer. This method simultaneously trains two RNN networks by minimizing a common loss function to obtain the final accurate position estimate [40]. Khassanov et al. proposed a positioning model that combines a four-layer neural network and long short-term memory (LSTM) to achieve end-to-end continuous indoor positioning [21].

From the research mentioned above, it can be observed that deep learning can be used for indoor positioning to improve positioning accuracy as illustrated in Figure 1. The core of deep learning is a neural network. Neural network algorithms mimic the structure and operation mechanism of brain neural networks, which are connected to a large number of neurons for information transmission [41]. Convolutional neural networks (CNN) contain multilevel nonlinear changes that are classical and widely used structures. The connection between neurons is inspired by the organization of the animal visual cortex [42]. CNNs have developed rapidly over the past decade and have the advantage of maintaining a strong feature extraction ability under a low number of parameters. Google, Facebook, and other business giants prioritized the research on CNN used in temporal modeling until TCN was first proposed by Bai et al. in 2018 [43]. A TCN is a type of special structure that can be used in the sequence of the convolution of the modeling problem of a neural network. It adopts a causal convolution mechanism, an expansion mechanism of convolution, and a residual connection. The TCN was not restricted by the size of the convolution kernel. It can be used to obtain a long history of information. Its structure is more compact, and its parameters can be significantly reduced. The TCN is suitable for solving traditional temporal gradient and gradient explosion problems of the network. Experiments have proven that TCN is superior to RNN and LSTM in terms of the efficiency and accuracy of temporal data analysis. For example, in the field of fault detection, Gao et al. proposed a TCN-based fault diagnosis model for power converters. The model has good robustness and reliability in noisy environments, and it can achieve adaptive feature extraction without any auxiliary pre-denoising algorithm. Compared to the LSTM-based diagnosis model, the fault recognition accuracy was improved by 6%. The number of parameters accounts for 1/4 of the LSTM diagnostic model. This model has advantages in terms of computational efficiency and time [44]. For the load forecasting problem of industrial customers, Wang et al. proposed a short-term load forecasting model for industrial customers based on the TCN and LightGBM(Light Gradient Boosting Machine). They applied a TCN to extract the deep characteristics and temporal relations of the input data. The model effectively reduced redundant features and improved the load-forecasting performance. Compared to the existing LSTM load forecasting model, the proposed model has better robustness and prediction effects [45]. 

In summary, research on indoor positioning has primarily focused on two-dimensional (2D) plane positioning, however, in practical application scenarios, most are multifloor environments, where 2D position coordinates cannot determine the location of the target, and it is necessary to obtain the vertical dimension information, floor, or height of the target in the building. In addition, the traditional location algorithm only considers the position matching of a single fingerprint and ignores the correlation between fingerprint sequences. The current position of the target is related to its historical position. Therefore, it is necessary to develop an indoor 3D positioning model that considers spatiotemporal data. Deep learning, which can be used in 3D positioning model design, has been an effective tool for data analysis in recent years.

## 3. Algorithm Design

This study proposed an indoor 3D positioning algorithm called TCN_DNNLoc, which is a hybrid algorithm combining TCN and DNN. The proposed algorithm is divided into an offline model training stage and an online fingerprint matching stage. The framework is illustrated in Figure 2.

In the offline stage, the temporal fingerprint map is constructed, the 3D positioning model is trained and the fingerprint data are collected in a continuous manner. WiFi fingerprints are collected along the trajectory and the corresponding 3D position coordinates of the collected points are recorded, along with the collected time stamp. Then, the fingerprint map is preprocessed and the RSSI in the fingerprint vector is converted to a number between 0 and 1 for improving the learning ability of the model. Next, a fixed-length sliding time window is used to construct temporal fingerprint slices, which are used as the input of the model, and the TCN feature extractor is used to learn the hidden information and temporal relationship in the input data. Finally, the extracted spatio-temporal features are transmitted to the DNN position regressor for nonlinear mapping of temporal features to 3D coordinates, and the training set is used for model training.

In the online fingerprint-matching stage, the positioning model is first loaded. The collected fingerprint is used to construct the timing fingerprint slice, which is input into the positioning model to realize end-to-end 3D position coordinate prediction. Finally, the positioning result is sent back to the user.

In the following sections, a temporal fingerprint map and its pre-processing method are introduced. Subsequently, a temporal fingerprint slice construction method based on a sliding window is discussed. Next, a fingerprint spatial and temporal feature-learning model based on the TCN is proposed, and finally, the proposed TCN_DNNLoc 3D temporal fingerprint positioning algorithm is introduced in detail.

### 3.1. Preprocessing of the Temporal Fingerprint Map

#### 3.1.1. Temporal Fingerprint Map

A 3D temporal fingerprint map is constructed continuously. The timestamp, WiFi fingerprint vector, and corresponding 3D position coordinates are recorded by continuous sampling along the trajectory at a certain time interval. The 3D temporal position fingerprint map data are listed in Table 2, with the fingerprints arranged in chronological order.

The WiFi fingerprint vector collected at time *t* can be expressed as shown in (1):(1)$$FP_{t}=\left\{MAC_{1}:RSSI_{1t},MAC_{2}:RSSI_{2t},\ldots ,MAC_{n}:RSSI_{nt}\right\}$$
where n represents the number of Access Points (Aps) in the environment and $MAC_{i}$ represents the *MAC* address of the *i*th AP in an environment. The *MAC* address is a unique AP identifier and $RSSI_{it}$ represents the Wi-Fi signal strength value from the *i*th AP at time *t*. The closer it is to the AP, the greater the RSSI value.

The temporal fingerprint map is defined as shown in (2):(2)$$\Omega =\{(FP_{1},Loc_{1}),(FP_{2},Loc_{2}),\ldots ,(FP_{m},Loc_{m})\}$$
where $FP_{t}$ represents the fingerprint vector collected at time *t* and $Loc_{t}$ is the 3D position vector corresponding to the sampling point at time *t*. The 3D position vector is defined as shown in Equation (3):(3)$$Loc_{t}=\{x_{t},y_{t},z_{t}\}$$

A temporal fingerprint-based indoor positioning problem can be formulated as a sequence learning problem that predicts the user’s 3D position vector $Loc_{t}$ at time *t* based on the current and previous fingerprint vectors, as shown in Equation (4).
(4)$$Loc_{t}=f(FP_{1},FP_{2},\ldots ,FP_{t})$$

Here, f(⋅) is a sequential function that allows exploiting temporal dependencies in the input.

#### 3.1.2. Temporal Fingerprint Map

This section describes the process of preprocessing WiFi RSSI. Through preprocessing, the RSSI value can be converted to a number between zero and one, which is helpful in improving the learning ability of the neural network model. Related studies have proved that WiFi RSSI preprocessing has a significant impact on positioning performance. Nurpeiissov et al. [46] proposed three representation methods for WiFi RSSI preprocessing, namely: zero-one normalized representation, exponential representation, and power representation, as shown in Equations (5)–(7), respectively.
(5)$$ZeroToOneNormalized(RSSI)=\frac{RSSI-c}{-c}$$
(6)$$Exponential(RSSI)=\frac{\exp(\frac{RSSI-c}{\alpha })}{\exp(\frac{-c}{\alpha })}$$
(7)$$Powed(RSSI)={\left(\frac{RSSI-c}{-c}\right)}^{\beta }$$
where the parameter *c* is a constant value that should be set to a number less than or equal to the minimum RSSI value to move the RSSI value to a positive range. The zero-one normalization representation performs a linear transformation of the original RSSI and normalizes the RSSI value after moving it to a positive range. The RSSI is more likely to have signal fluctuations when it is far from the AP. Consequently, the exponential representation and power representation perform nonlinear transformations on the RSSI and add penalties for higher RSSI values. The denominator parameter α is set to a constant and the exponential parameter *β* is generally set to a mathematical constant *e*. After preprocessing, the RSSI value was converted to a value between 0 and 1. An RSSI value close to 1 indicates a stronger signal and 0 indicates an undetected AP.

Preprocessing of the WiFi RSSI has a significant influence on positioning accuracy. For the IMUWIFINE dataset [39] used in this study, the positioning situations of zero-one normalization representation, exponential representation, power representation, and no preprocessing were tested. The RSSI value in the dataset ranges from −93 dBm to −34 dBm and the RSSI value of the undetected AP was set to −100 dBm; thus, *c* was set to −100 in this study. The experimental results are shown in Figure 3. It can be observed that WiFi RSSI preprocessing significantly improves the positioning effect. In addition, the power representation has the best effect, and consequently, a power representation is chosen to conduct the RSSI data preprocessing. 

### 3.2. Construction of Fingerprint Slice

Traditional positioning algorithms use a single fingerprint as the model input. Owing to the noise and redundancy in WiFi fingerprints, fingerprints collected for the same location may not be the same. Additionally, once the points that are far away from each other in the physical space have a high similarity in the signal space, it will affect the positioning accuracy. A sliding time window to build a temporal fingerprint chip as the input of the positioning model to provide abundant information for positioning. Temporal information can be used to distinguish locations with similar fingerprint vectors and to improve the accuracy and robustness of positioning.

In this study, a fingerprint vector sequence is selected as the input of the model. Temporal fingerprint slices are constructed by window sliding, where the size of the sliding window was set as W, and the fingerprint vectors in continuous-time t−W+1 to time were set t as a group to form a temporal fingerprint slice. Each temporal fingerprint slice contains W fingerprint vectors collected continuously. Fingerprint vectors have spatiotemporal dependence and the positioning accuracy can be improved by combining the historical fingerprint information. Considering the size of the window W=4 as an example, the construction method of the sliding window is introduced, as shown in Figure 4. A window of size four was maintained, and a step of one is taken to continuously slide down on the temporal fingerprint dataset. The fingerprint vector collected from $t_{1}$ to $t_{4}$ constitutes the fingerprint slice and the fingerprint vector collected from $t_{2}$–$t_{5}$ constitutes another fingerprint slice. Fingerprinted slices are constructed successively in this manner.

The reconstructed fingerprint slice dataset using the sliding window is shown in Table 3. Each fingerprint slice contains four fingerprint vectors collected continuously, all of which contain time information. The position coordinate corresponding to the last sample in the window is taken as the position label of the entire fingerprint slice.

### 3.3. Positioning Model Design

The model architecture is shown in Figure 5. It is primarily divided into four modules, namely: a fingerprint slice input module, a TCN feature extractor module with three residual blocks, a DNN regressor module with two hidden layers, and a 3D coordinate output module.

#### 3.3.1. Input Module

The fingerprint vector of a single strip is easily affected by the dynamic changes in the environment and the inherent noise of the signal; therefore, the fingerprint slice *Series_FP* = [*FP_t_*_-*w*+1_, *FP_t_*_-*w*+2_, …, *FP_t_*] is used as the input of the model. *Series_FP* is a two-dimensional matrix of W × N. W indicates the sequence length of time, which is equal to the size of the sliding window. Here, N indicates the total number of AP in the environment. The fingerprint slice contains a fingerprint vector collected continuously over time, W. The location label corresponding to the fingerprint slice is ${\mathrm{Loc}}_{t}=\{x_{t},y_{t},z_{t}\}$, which represents the actual physical location coordinates of the sampling point at time *t*.

#### 3.3.2. TCN Feature Extractor Module

The TCN feature extractor with three residual blocks is used to extract the features of *Series_FP*, as shown in Equation (8) and ST_Features indicates the extracted spatio-temporal features.
(8)ST_Features=TCN(Series_FP)

The specific architecture of residual blocks is shown in Figure 6.

Each residual block contained two layers, namely: causal and dilated convolutions. Through the causal convolution mechanism, causal constraints are added, and the connections that do not meet the temporal dependence are removed in the training and learning processes. Through the dilated convolution mechanism, larger receptive fields of information are obtained without increasing the number of network layers. The weight-normalized layer is added after each causal and dilated convolution layer to speed up model convergence. The ReLU activation layer activates neurons nonlinearly and adds nonlinear factors to the model to improve its feature learning ability. The Droupout layer sets the loss rate to inactivate neurons proportionally and at random to avoid an excessive fitting model. This is discussed in detail in the next section.

The framework of the TCN feature extractor is shown in Figure 7. where, d denotes the dilation coefficient. The dilation coefficient of the three residual blocks increases exponentially, which causes the information receptive field to expand by two times, in turn. Here, k represents the size of the convolution kernel used for feature extraction. The frameworks of the three residual blocks are identical, except for the expansion coefficient. The residual connection mechanism prevents network degradation, which effectively solves the problems of gradient disappearance and explosion.

#### 3.3.3. DNN Regressor Module

After extracting spatio-temporal features, the TCN feature extractor is connected to the DNN regressor with two hidden layers to learn the nonlinear mapping between spatio-temporal features and 3D positions, as shown in Figure 6. To prevent network overfitting, a dropout layer is added after each hidden layer. The dropout layer is composed of three neuron outputs for the 3D position coordinate estimation.

The *ST_Features* vector is passed forward to the DNN regressor with two hidden layers, as shown in Equations (9)–(12):(9)$$y_{1}=f(w_{1}ST\_Features+b_{1})$$
(10)$$y_{1}^{\prime }=drop(y_{1})$$
(11)$$y_{2}=f(w_{2}y_{1}^{\prime }+b_{2})$$
(12)$$y_{2}^{\prime }=drop(y_{2})$$

Here, $w_{1}$ and $w_{2}$ are the weights of the two fully connected hidden layers, whilst $b_{1}$ and $b_{2}$ are the biases of the two hidden layers. The variables $y_{1}$ and $y_{2}$ are the outputs of the two hidden layers and f(⋅) indicates the activation function. ReLU activation function is choosen for the model, which adds nonlinear factors to the model, converts the linear result into a nonlinear output, improves the expression ability of the neural network to the model, and solves the problem that the linear function cannot be fitted. The use of the ReLU activation function and dropout layer can alleviate the overfitting problem to some extent.

#### 3.3.4. Output Module

After passing through two fully connected hidden layers, information is transmitted to the output layer with three neurons, which do not use the activation function, as shown in Equation (13):(13)$$o_{i}=w_{3i}y_{2}^{\prime }+b_{3i},i=1,2,3$$

The $o_{i}$ represents the value of each neuron in the output layer, $o_{1}$ represents the x-coordinate predicted by the model, $o_{2}$ represents the Y-coordinate predicted by the model and $o_{3}$ is the Z coordinate predicted by the model. The variables $w_{3i}$ and $b_{3i}$ represent the connection weight and bias, respectively, from the second hidden layer to the ith neuron in the output layer.

The mean square error was used as the loss function of the model, as shown in Equation (14):(14)$$L_{MSE}(Loc_{t},\hat{Loc_{t}})=\frac{1}{T}\sum_{t=0}^{T}||Loc_{t}-\hat{Loc_{t}}|{|}_{2}$$
where $Loc_{t}$ denotes the actual 3D coordinates at time *t*, $\hat{Loc_{t}}$ denotes the 3D coordinates predicted by the model at time *t*. An Adam optimization was used to update the parameters.

## 4. Experiments and Analysis

### 4.1. Dataset

The experiment used the IMUWIFINE dataset [46] launched by ISSAI, which is an open-source position fingerprint dataset with temporal information. The collection site was located in Building C4 on the campus of Nazarbaye University. It covers a total area of over 9000 square meters on three floors: the fourth, fifth, and sixth. The fingerprint vector and corresponding 3D position coordinates were recorded every five seconds along a continuous track. A collection of examples is presented in Figure 8.

Specific attribute information for the dataset is presented in Table 4. A total of 220 APs were placed in the environment and a default value of −100 dBm was used to represent the undetected APs.

### 4.2. Evaluation Index

The 3D average error distance and cumulative error distribution functions were used as the performance evaluation indexes of the model. 

The definition of 3D average error distance is shown in Equation (15):(15)$$3D\_MED=\frac{\sum_{i=1}^{N}dist(Loc_{i},\hat{Loc_{i}})}{N}$$
where $Loc_{i}$ is the actual 3D position coordinate of the target predicted by the positioning model introduced in Section 3.3. *dist*() is the Euclidean distance between the actual position $Loc_{i}$ and predicted position $\hat{Loc_{i}}$, whilst N is the sum of the number of samples in all tests.

The *dist*() is defined as in Equation (16):(16)$$dist(Loc,\hat{Loc})=\sqrt{{\left(x-\hat{x}\right)}^{2}+{\left(y-\hat{y}\right)}^{2}+{\left(z-\hat{z}\right)}^{2}}$$

The cumulative error distribution function (CEDF) is another evaluation index and the trend in the error distribution curve indicates the overall error distribution. The faster the curve rises, the better the positioning effect.

### 4.3. Parameter Setting

To improve the performance of the positioning model, the TCN feature extractor and DNN position regressor were optimized.

The number of convolution kernels in the TCN feature extractor affects the spatiotemporal feature extraction, which in turn affects the accuracy of the model. This study tested the effect of the TCN with different numbers of convolution kernels. The experimental comparison results are shown in Figure 9.

Initially, with an increase in the number of convolution kernels, the positioning precision of the model was greatly improved. The best positioning effect was obtained by using 128 convolution kernels. As the number of convolution kernels continued to increase, the model positioning performance decreased. Additionally, when the number of model parameters increased dramatically, the training time became significantly longer. Consequently, the number of convolution kernels in the TCN space-time feature extractor was set to 128.

For DNN regressors, different network structures affected the performance of the model. The experimental comparison results are shown in Figure 10.

It can be observed that the performance of a single hidden layer was limited. However, increasing the number of network layers can cause overfitting and affect the positioning performance of the model. In the experiment, 128–128 means that two hidden layer structures with 128 neurons were adopted; this selection yielded the best positioning performance. 

This study adopted a sliding window to build the temporal fingerprint slice where the length of the temporal fingerprint slice was equal to the size of the sliding window. The longer the series, the more historical information it contains but the more cumulative errors it introduces. To balance these two points for the optimal time series length, experiments were conducted with different series lengths. The positioning errors for the different series lengths are shown in Figure 11.

The minimum average error distance was obtained when the time-series length was eight. The cumulative error distribution curves for different time-series lengths are shown in Figure 12. It can be observed from the figure that when W = 8, the error distribution curve rises the fastest, and the curve is the shakiest. More than 80% of the positioning results achieved errors of less than two meters. The details can be found in Table 5. 

In summary, parameter settings of TCN_DNNLoc algorithm are shown in Table 6. 

### 4.4. Performance Comparison of Different Algorithms

The performance of the proposed TCN_DNNLoc model was compared with the nonsequential model SRL_KNN [39], FNN [21], and sequential model LSTM [46]. All hyperparameters were tuned on the validation set, and the final model was evaluated on the test set of IMUWiFine Dataset. For SRL_KNN, K is set to 3. For FNN, the model is with 5 hidden layers and 128 hidden units. ReLU is used in all hidden layers. More details about the model can be found in Ref. [21]. For LSTM, the model is consisting of four ReLU and three LSTM layers with a dimension size of 128 units. More details about the model can be found in Ref. [46]. The epochs of all the model are set the same as 1000. 

The 3D positioning errors of different models are shown in Figure 13.

It can be observed from the figure that the positioning accuracy of the sequence model is significantly higher than that of the nonsequence positioning model. For the comparison between the two sequential models, the 3D average error distance of the positioning model based on LSTM was 1.33 m, and the 3D average error distance of the TCN_DNNLoc was 1.22 m, an accuracy improvement of 7.5%. The cumulative error distribution curves for the different models are shown in Figure 14.

It can be observed from Figure 14 that the error distribution curve of TCN_DNNLoc increased the fastest, and the curve was the shakiest. More than 80% of the positioning results achieved positioning errors of less than two meters, and the positioning performance was the best, proving the validity of the TCN_DNNLoc model. The details are included in Table 7.

## 5. Conclusions

In our daily life, 3D indoor positioning is of great significance. It can not only help people navigate in multi-story buildings, but also help managers to count the distribution of people flow in the buildings. At present, it can also help to control the spread of COVID-19 more accurately. This study reviewed relevant references and comprehensively analyzed existing positioning techniques and methods. We focus on the WiFi fingerprint method because of its wide deployment in multi-story buildings. Compared to a single fingerprint input, the fingerprint sequence collected continuously contains historical data, which can provide richer information for positioning. Temporal information can be used to distinguish positions with similar fingerprint vectors and improve the accuracy and robustness of mobile positioning. Based on previous studies, this study applied deep learning techniques to the indoor positioning field and proposed a new sequence-positioning model called TCN_DNNLoc. 

In the proposed model, the sliding time window method is used to construct temporal fingerprint slices, which are used as the input to the model, to provide richer information for positioning, solve the problems of environmental noise and spatial ambiguity, and improve the accuracy and robustness of positioning. A TCN spatiotemporal feature extractor was proposed to extract the sequence fingerprint slice of hidden information and temporal relations. The proposed TCN model adopts causal convolution, dilated convolution, and residual connection mechanisms, and is not restricted by the size of the convolution kernels. The proposed model also expands the receptive with long history information, compact structure, and reduced parameters, and solves the problems of gradient disappearance and gradient explosion. The extracted spatiotemporal features are connected to a deep neural network (DNN) position regressor, and the complex nonlinear mapping relationship between the features and 3D position coordinates is learned to estimate the 3D position coordinates of the target. The effectiveness of the TCN_DNNLoc model was proved by an experimental comparison with the existing 3D method.

There are two future research directions of this study:(1)The positioning accuracy was affected by the connection weight and bias of the model. Future research should consider using a heuristic algorithm to optimize the parameters. Examples include the genetic algorithm (GA) [47], particle swarm optimization (PSO) [48], simulated annealing [49], and quantum annealing algorithms [50]. The optimization algorithms can be used to optimize the connection weight matrix and bias of the positioning model to avoid falling into the local optimal solution. Moreover, the convergence speed of the model can be improved.(2)This study considered only WiFi technology for positioning; however, each positioning technology has its own characteristics and limitations. With the development of microelectric systems, the accuracy of sensors has improved continuously, whilst the built-in sensors of mobile phones, such as air-pressure sensors and accelerometers, can be used for auxiliary positioning. Future research must consider the integration of WiFi technology with other technologies to provide more accurate positioning information. Multi-technology fusion and mutual learning can compensate for the limitations of single-positioning technology and improve positioning accuracy.

## Figures and Tables

**Figure 1 sensors-23-00153-f001:**
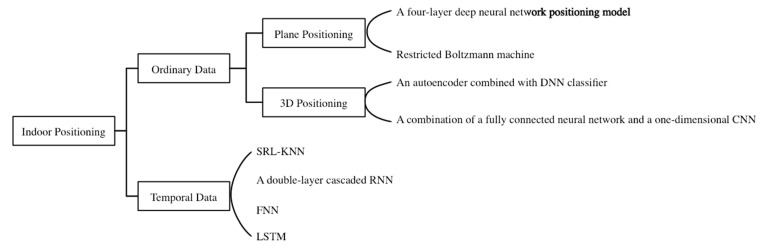
Some Deep Learning Methods in WiFi fingerprint-based Indoor Positioning.

**Figure 2 sensors-23-00153-f002:**
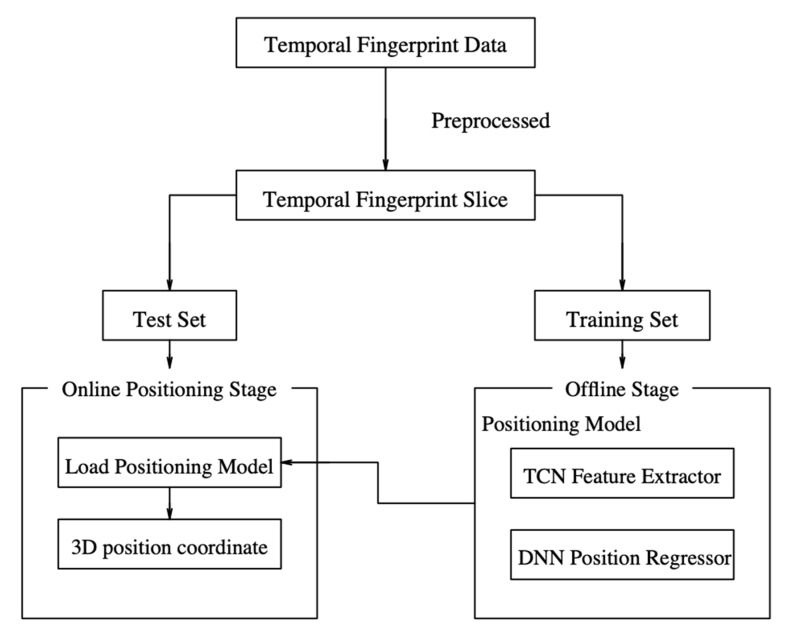
TCN_DNNLoc framework.

**Figure 3 sensors-23-00153-f003:**
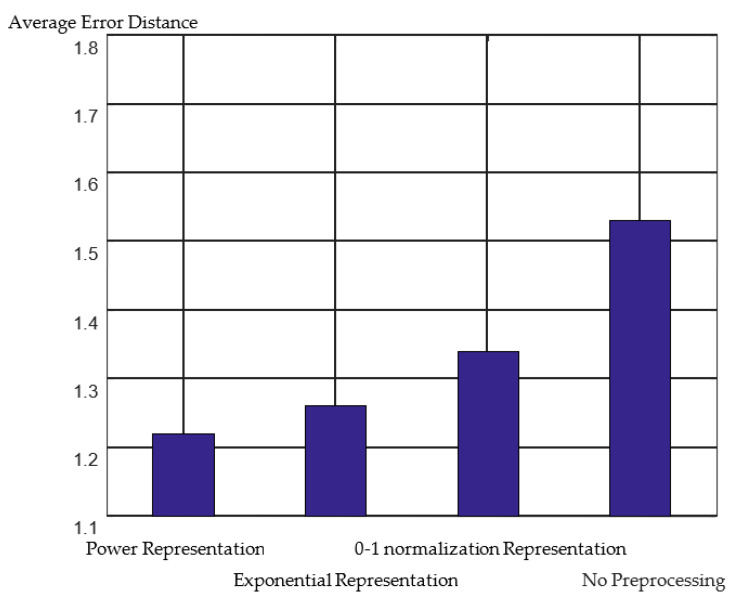
3D positioning errors of different preprocessing methods.

**Figure 4 sensors-23-00153-f004:**
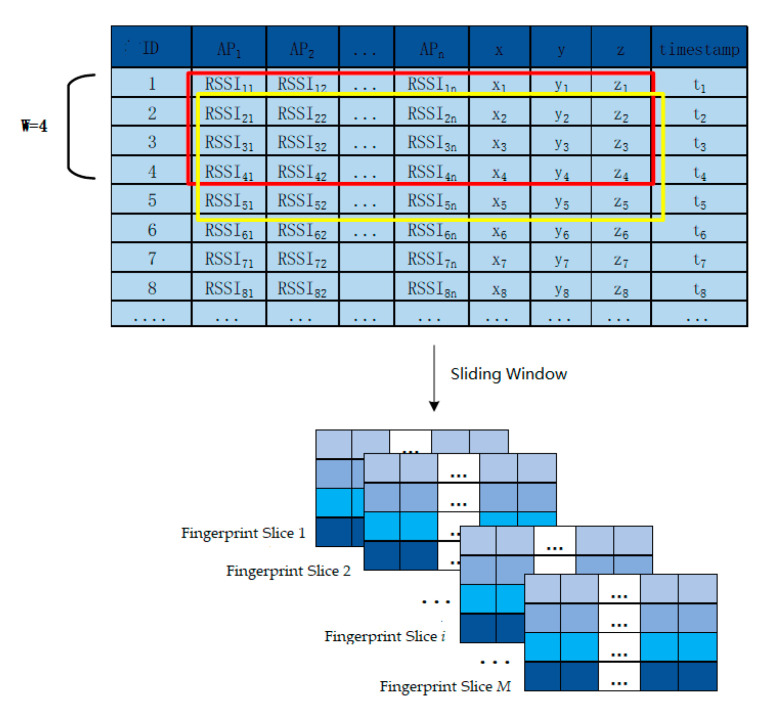
Fingerprint slice construction process.

**Figure 5 sensors-23-00153-f005:**
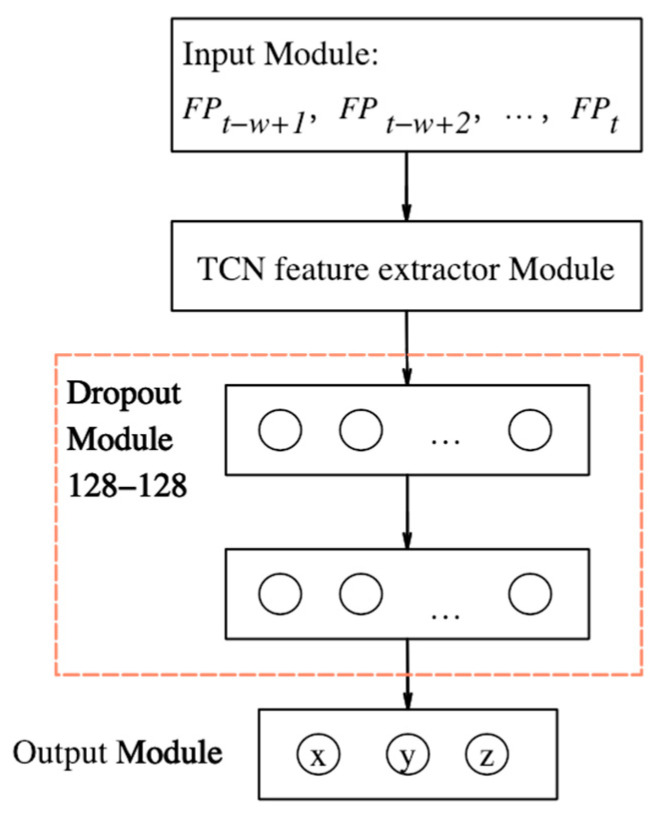
Positioning Model Architecture.

**Figure 6 sensors-23-00153-f006:**
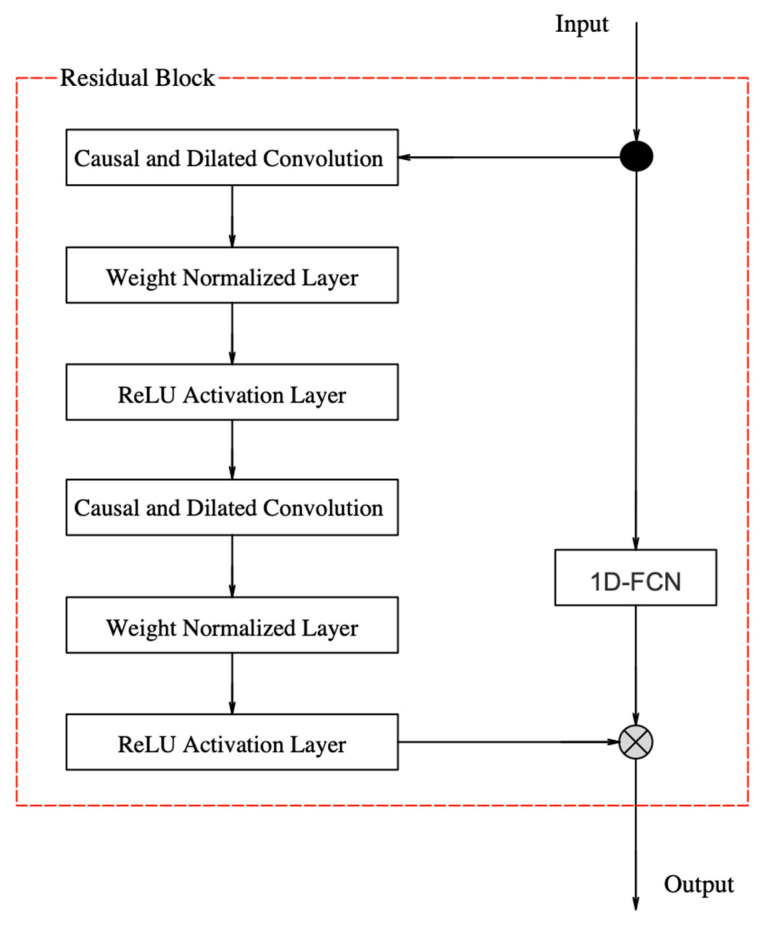
TCN residual block architecture.

**Figure 7 sensors-23-00153-f007:**
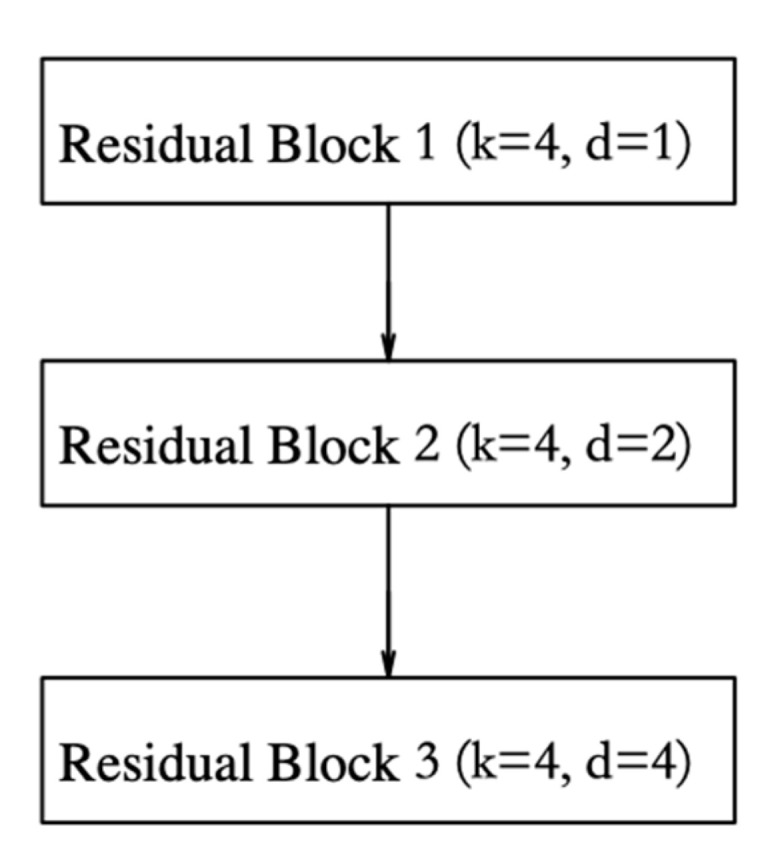
TCN Feature Extractor.

**Figure 8 sensors-23-00153-f008:**
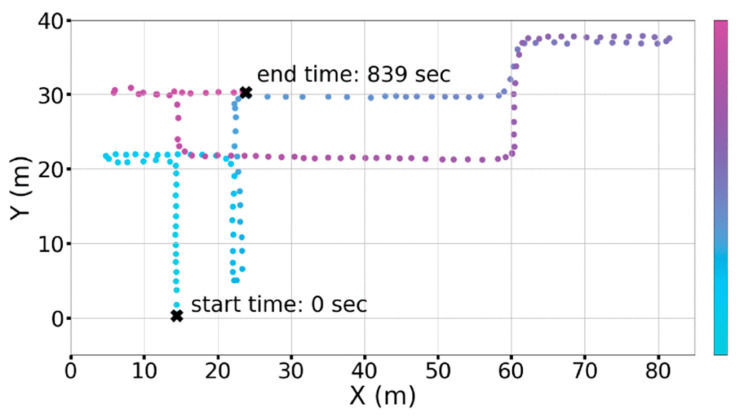
An example of temporal fingerprint collection.

**Figure 9 sensors-23-00153-f009:**
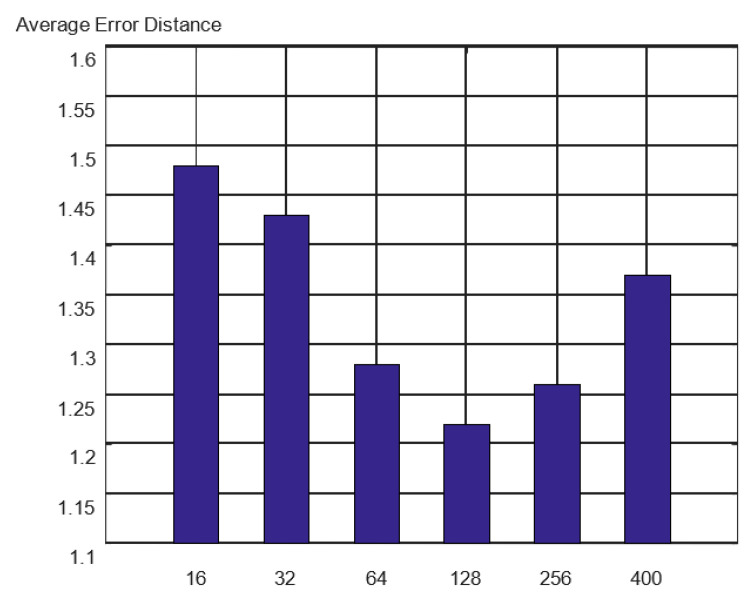
Localization effect of different convolution kernel numbers.

**Figure 10 sensors-23-00153-f010:**
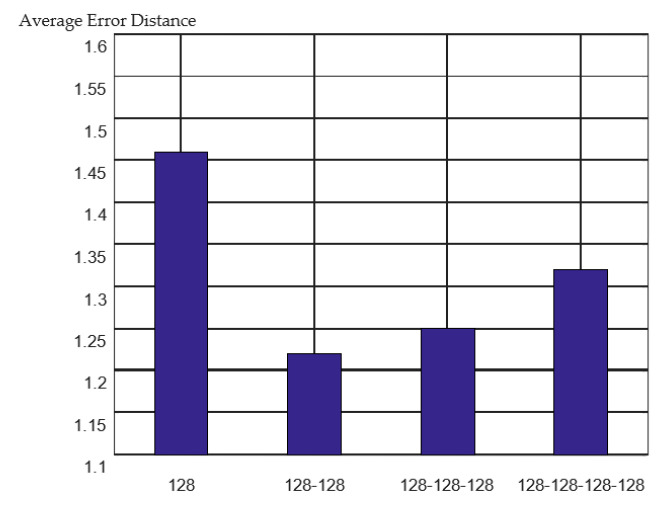
Localization effect of different DNN structures.

**Figure 11 sensors-23-00153-f011:**
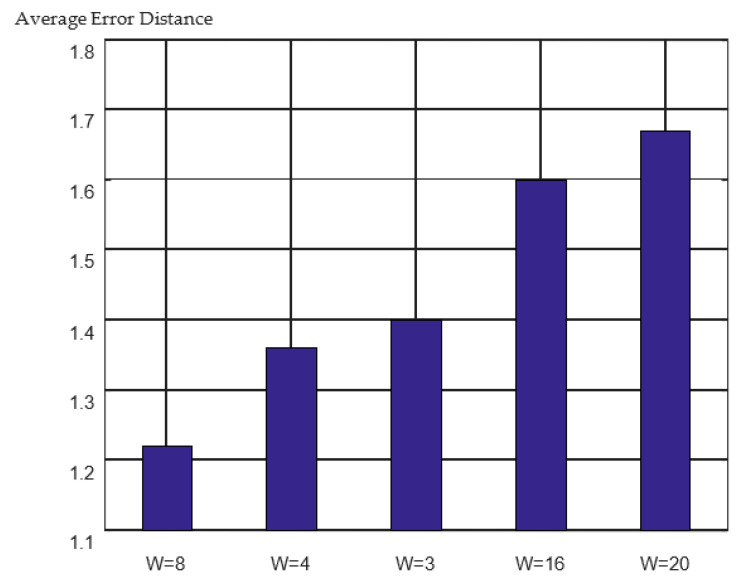
3D positioning error of different series length.

**Figure 12 sensors-23-00153-f012:**
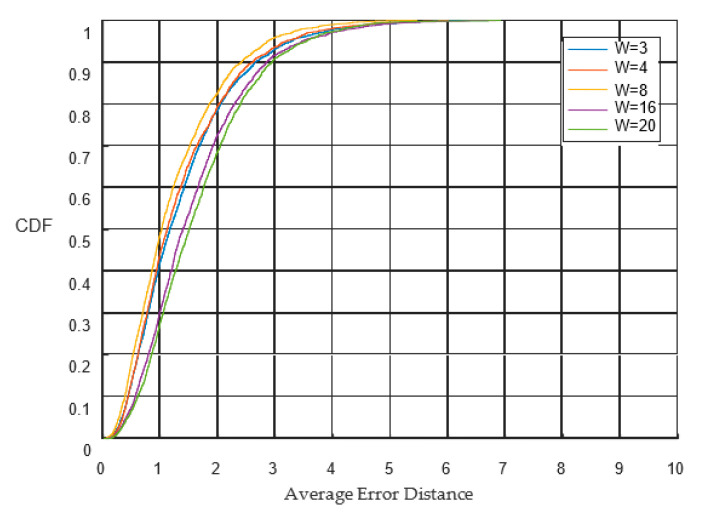
Cumulative distribution functions for different sequence lengths.

**Figure 13 sensors-23-00153-f013:**
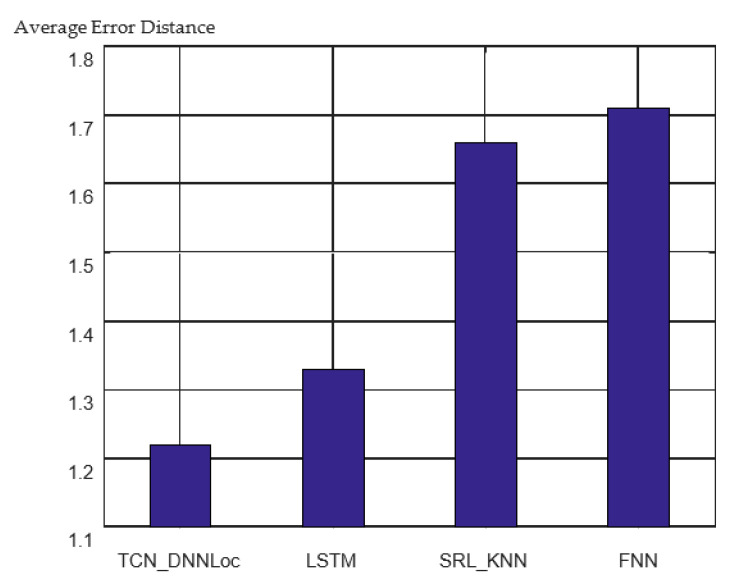
3D positioning error of different models.

**Figure 14 sensors-23-00153-f014:**
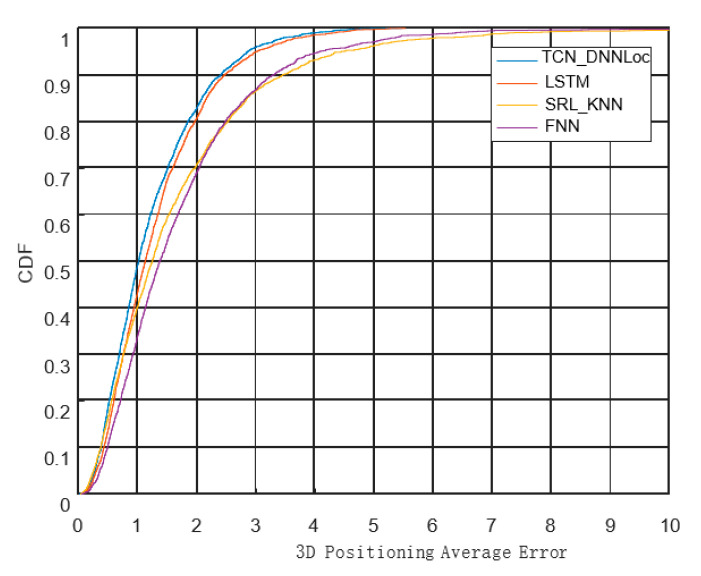
Cumulative distribution functions for different models.

**Table 1 sensors-23-00153-t001:** Major indoor positioning technologies.

Positioning Technique	Advantage	Disadvantage	Positioning Accuracy	Popular System
Ultrasonic [6]	High positioning accuracy	Multipath effect, thermal drift effect, high cost, severe decay	Centimeter scale	Bat [7]
RFID [8]	Low cost, high speed	Short distance, low communication capacity	Centimeter scale	LANDMARC [9]
UWB [10]	Strong resistance to interference, strong penetration, high positioning accuracy	High cost	Sub-meter scale	Dart UWB [11]
Bluetooth [12]	Low power consumption, easily deployable	Short distance	Meter scale	BIPS [13]
Infrared [14]	Low-cost, mature technology	Poor resistance to interference	Meter scale	Active Badge [15]
Zigbee [16]	Low-cost, low power consumption	Poor stability	Meter scale	[17]
WiFi [18]	No additional hardware required for deployment, low cost, wide application range	Tedious fingerprint collection, degeneration of WiFi signal	Meter scale	RADAR [19]

**Table 2 sensors-23-00153-t002:** 3D temporal fingerprint map data.

Fingerprint ID	AP_1_	AP_2_	...	AP_n_	x	y	z	Timestamp
1	RSSI_11_	RSSI_12_	...	RSSI_1n_	x_1_	y_1_	z_1_	t_1_
2	RSSI_21_	RSSI_22_	...	RSSI_2n_	x_2_	y_2_	z_2_	t_2_
3	RSSI_31_	RSSI_32_	...	RSSI_3n_	x_3_	y_3_	z_3_	t_3_
4	RSSI_41_	RSSI_42_	...	RSSI_4n_	x_4_	y_4_	z_4_	t_4_
5	RSSI_51_	RSSI_52_	...	RSSI_5n_	x_5_	y_5_	z_5_	t_5_
6	RSSI_61_	RSSI_62_	...	RSSI_6n_	x_6_	y_6_	z_6_	t_6_
....	...	...	...	...	...	...	...	...

**Table 3 sensors-23-00153-t003:** Fingerprint slice dataset.

Fingerprint Slice	Position Coordinate
[FP_1_, FP_2_, FP_3_, FP_4_]	x_4_, y_4_, z_4_
[FP_2_, FP_3_, FP_4_, FP_5_]	x_5_, y_5_, z_5_
[FP_3_, FP_4_, FP_5_, FP_6_]	x_6_, y_6_, z_6_
…	…
[FP_M_, FP_M+1_, FP_M+2_, FP_M+3_]	x_M+3_, y_M+3_, z_M+3_

**Table 4 sensors-23-00153-t004:** IMUWIFINE dataset details.

Attribute	Meaning
AP001-AP220	RSSI from the corresponding AP (dbm)
x	Value on the X-coordinate (m)
y	Value on the Y-coordinate (m)
z	Value on the Z-coordinate (m)
Timestamp	Timestamp of the sample (s)

**Table 5 sensors-23-00153-t005:** The proportion of the average error distance for different time-series lengths.

	<1 m	<2 m	<4 m	<6 m
W = 3	41%	79%	98%	100%
W = 4	45%	79%	98%	100%
**W = 8**	**49%**	**82%**	**99%**	**100%**
W = 16	30%	73%	97%	100%
W = 20	28%	69%	97%	100%

**Table 6 sensors-23-00153-t006:** TCN_DNNLoc model parameters.

Parameter	Value
Epochs	1000
Doupout	0.1
Residual blocks	3
Convolution kernel size	4
Number of convolution kernels	128
activation function	ReLU
Loss function	Mse
Optimization algorithm	Adam

**Table 7 sensors-23-00153-t007:** The proportion of the average error distance of different models.

	<1 m	<2 m	<4 m	<6 m	Average Error (m)	
					Valid	Test
TCN_DNNLoc	**49%**	**82%**	**99%**	**100%**	**1.21**	**1.22**
LSTM	42%	81%	98%	100%	1.35	1.33
SRL_KNN	40%	70%	93%	96%	1.62	1.66
FNN	34%	69%	94%	97%	1.75	1.71

## Data Availability

The data presented in this study are openly available in [46].

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
