# Peer review of "Research on Indoor 3D Positioning Algorithm Based on WiFi Fingerprint"

_sensors, 2022, doi:10.3390/s23010153_

Round 1

Reviewer 1 Report

The paper proposes integration of a deep learning algorithm for indoor 3D positioning and a 3D dynamic positioning model based on temporal fingerprints which uses a sliding time window to build a temporal 13 fingerprint chip as the input of the positioning model. Deep learning has been applied for the spatiotemporal feature extraction of 17 features, which are then connected with a deep neural network (DNN) to estimate the 3D position coordinates of the target. Additionally, a public dataset is used to verify the performance of the localization algorithm. Authors claim that the proposed model has better accuracy that the existing models for estimating three-dimensional position coordinates.

Some improvements suggested:

1.      The architecture of the deep learning model should be explained in detail with proper justification for selection of parameters and hyperparameters like number of layers, nodes, learning parameters, etc.

2.      How does proposed model remove ‘spatial ambiguity’ and increase ‘robustness’ as claimed by authors? This should be explained.

3.      Applications of such a proposed system should be mentioned for the wider audience.

Reviewer 2 Report

The authors proposed a novel deep-learning method for 3D positioning. The manuscript is generally well-written and structured, but there are still some concerns on the method.

Major comments:

  1. There are many deep learning 3D positioning research efforts, it is better to put some efforts on summarizing these results in a table like Table 1 in the literature search section. For example, https://mdpi-res.com/d_attachment/sensors/sensors-20-05495/article_deploy/sensors-20-05495.pdf?version=1601024305

  2. For study design, it seems that there is no cross validation. Also, there is not training validation and testing split for the analysis. This might lead to over estimation of the algorithm performance. It would be better to use separate validation set for parameter tuning and include the cross validation results.

  3. Please include the tuning details for other competing methods. Especially for LSTM and FNN.

  4. What’s the current SOTA methods for deep learning 3D positioning? It seems that the competing methods are baseline models.

  5. For figure 13, please translate the Chinese character.

  6. For CDF results, if would be better to include necessary statistical test results for curve comparisons.

Reviewer 3 Report

This paper explores indoor 3D positioning using wireless access points and deep learning. 3D positioning is an essential issue and the paper has given a feasible solution. Overall, the paper is well-written with good logic, and can be accepted as long as the following issues are addressed:

1. The authors are suggested to conduct careful proofreading to correct the minor mistakes, e.g., the caption of Section 3.3.

2. In Fig. 5, the activation function should read ‘ReLU’ instead of ‘Relu’. Besides, the ‘dropout’ is only conducted in the training process, and is not suggested to be listed as a layer in the network architecture.

3. In Section 4.3, Figs. 8 and 9 shows the effect of convolution kernel numbers and DNN layers on the positioning accuracy. Have the authors used the same number of training episodes? Theoretically, a larger convolution kernel or a deeper neural network may need more training episodes since the number of trainable parameters has increased.

4. In Fig. 13, please revise the Chinese characters into English.

5. Similar to comment 3, a fair comparison may be ensured by providing the number of trainable parameters in Section 4.4, as currently the authors have not shown any detail of other models.

(Optional, the authors may choose not to answer this question) Besides, the reviewer is curious if the current WiFi fingerprint-based approach can be used to the positioning task under fire conditions. If so, this technology can also benefit firemen so that they do not get lost at fire rescue scenes.

Author Response

Thanks for the comments and suggestions. Please kindly find the attached file. 

Round 2

Reviewer 2 Report

Thank the authors for the replies.

Figure 12 and Figure 13 still include Chinese characters.

For the study design, the authors claimed that they have performed cross-validation for model training. If that is the case, the performances of cross-validation for all the methods should be reported.  

From Line 478 to Line 484, there are no tuning details for competing methods.

Author Response

Please kindly find the attachment.  Sorry, I uploaded a wrong manuscript version before. Sorry for any inconvenience. 
